# Bacilli in the International Space Station

**DOI:** 10.3390/microorganisms10122309

**Published:** 2022-11-22

**Authors:** Andrea Quagliariello, Angela Cirigliano, Teresa Rinaldi

**Affiliations:** 1Department of Comparative Biomedicine and Food Science, University of Padova, 35122 Padova, Italy; 2Istituto di Biologia e Patologia Molecolari—IBPM CNR, 00185 Rome, Italy; 3Department of Biology and Biotechnology, Sapienza University of Rome, 00185 Rome, Italy

**Keywords:** *Bacillus*, ISS, *B. anthracis*, *B. cereus*, evolution

## Abstract

Astronauts remote from Earth, not least those who will inhabit the Moon or Mars, are vulnerable to disease due to their reduced immunity, isolation from clinical support, and the disconnect from any buffering capacity provided by the Earth. Here, we explore potential risks for astronaut health, focusing on key aspects of the biology of *Bacillus anthracis* and other anthrax-like bacilli. We examine aspects of *Bacillus cereus* group genetics in relation to their evolutionary biology and pathogenicity; a new clade of the *Bacillus cereus* group, close related to *B. anthracis*, has colonized the International Space Station (ISS), is still present, and could in theory at least acquire pathogenic plasmids from the other *B. cereus* group strains. The main finding is that the genomic sequence alignments of the *B. cereus* group ISS strains revealed a high sequence identity, indicating they originated from the same strain and that a close look to the genetic variations among the strains suggesting they lived, or they are living, in a vegetative form in the ISS enough time to accumulate genetic variations unique for each single strains.

## 1. Introduction

Human populations, like those of most other animal species, are vulnerable to outbreaks of infectious disease. Given the uniqueness of our Earth, with its rich biosphere and conditions to which human biology has evolved and is adapted, it is an understatement to say that the entire human population could not survive on a hostile, likely sterile, and unproductive Mars. There has not yet been a human mission to Mars, but the condition of the human system is known to deteriorate outside Earth. Studies of astronaut health indicate that these include a loss of bone mass, potential muscle wastage, risk to eyesight, reduced immunity, stressful conditions [1,2,3,4], and perturbations in the microbiome [5,6,7]. Astronauts are at the same time reliant on a connection with terrestrial support and isolated from direct clinical back-up, so are particularly vulnerable to disease. This is likely exacerbated by the disconnect from any buffering capacity provided by the terrestrial ecology. This can potentially lead to the emergence of pathogenic microbiomes within human habitations, whether in transit of on a planetary surface, such as that of Mars. The current article focuses on anthrax-causing *Bacillus* as a lens through which to investigate this risk.

*Bacillus* species can be found virtually everywhere and have the capacity of survival in many habitats, reflecting the genetic polymorphism of this bacterial genus. The *B. cereus* group (which comprises *B. cereus*, *B. thuringiensis*, *B. anthracis*, *B. weihenstephanensis*, *B. mycoides*, *B. pseudomycoides* and *B. cytotoxicus* species), grown in the soil environment with a saprophytic life cycle, as symbiotic with plant roots or guts of insects and mammals [8,9], and they could survive for long periods as spores [10]. The spores and biofilms confer to the *B. cereus* group a high resistance to various stresses and a high adhesive capacity on every substrate [11], including stainless steel, underlining a problem on the control of surface contamination [12]. Indeed, strains of *B. cereus* are also frequent agent of food poisoning outbreaks [13,14] and could act as spoilage organisms in the food industry [15]. In isolated systems, such as the ISS, spacecrafts, or planetary habitats, the biofilm production on surfaces or materials is a concern because it is known that biofilm production increases bacterial resistance to antibiotics [16].

Nomenclature of micro-organisms aims to classify down to the species level, but this is challenging for some members of the *Bacillus cereus* group:*Bacillus anthracis* is found in soil and it causes the anthrax disease in ungulates;*Bacillus cereus* that is a soil inhabitant, can be isolated from foods, such as grains and spices, and can occasionally cause food-borne intoxications in humans;*Bacillus thuringiensis* is distinguished from *B. cereus* or *B. anthracis* by its pathogenicity for Lepidopteran insects. The different members of this group share chromosomal features, with few differences to distinguish them (i.e., loss of function mutation of the PlcR gene in the *B. anthracis* strains), so are essentially distinguished by species-specific plasmids; as an example, the pathogenicity of *B. anthracis* relies on the presence of both the pOX1 and pOX2 plasmids, pXO1 codes for the tripartite toxin and pXO2 codes for the polyglutamate capsule, needed to circumvent the immune system avoiding the phagocytosis of macrophages [17]. Anthrax is a disease observed in animals (ungulates); humans become infected only incidentally if they come into contact with infected animals. The most common form of the disease in humans is cutaneous anthrax, acquired when *B. anthracis* (or its spores) comes in contact with injured skin; in the site the spores germinate, bacteria multiply, and a gelatinous edema develops and results into a necrotic ulcer from which infection may disseminate. The intestinal anthrax is caused by poorly cooked meat ingestion of infected animals. Another form of the disease, inhalation anthrax, results from inhalation of spore-containing dust where animal hair or hides are present [18], the *Bacillus anthracis* spores survive for many years [10].

Because of the close relation between *B. anthracis, B. cereus*, *B*. *thuringiensis,* it has been proposed that *B. anthracis* is a lineage/subspecies of *B. cereus* [19]. These species also undergo to horizontal gene transfer and plasmid exchange [20,21]. In their primary habitat, the soil, the members of the *B. cereus* group are known to exchange plasmids [22,23]. A non-pathogenic strain which acquires a pathogenic plasmid can be transformed in a potential pathogen. By way of example, patients infected by *B. cereus* strains harboring a plasmid virtually identical to the *B. anthracis* pXO1 plasmid, which code for the tripartite (anthrax) toxin, presented with an inhalation anthrax-like illness [24,25,26,27,28,29,30,31,32].

*B. anthracis* is included in the list of micro-organisms that have the potential to pose a severe threat to public health and that were produced as bioweapons in the past (its spores are used to cause inhalation anthrax) [33]. This group of micro-organisms are also named biological agents, and their use, including the basic research, is scrutinized by governmental agencies to prevent the illegal proliferation and use of such organisms and also to ensure compliance with the Biological and Toxins Weapons Convention and the United Nations Security Council Resolution 1540 [34]. Whereas *B. anthracis* is quite rightly a focus of attention in the field of biosecurity, the dangers posed by *B. cereus* strains are generally overlooked. The examples of human anthrax-like diseases caused by pathogenic *B. cereus* strains remain accidental and rare cases, but the increasing scientific knowledge on these strains (plasmid shuffling and horizontal gene transfer) and the low cost of DNA sequencing could accelerate the discovery of other pathogenic strains in a near future.

The 2010 discovery of a new strain called *Bacillus cereus* biovar *anthracis* strain in Africa [35] has complicated the phylogenetic classification of the *B. cereus* group. The species designation is *B. cereus*, but its pathogenicity reveals that it causes an anthrax-like disease in chimpanzees and gorillas in rainforest habitats [36,37]. It has also a different ecology than *B. anthracis* which usually causes anthrax in herbivores (ungulates) that inhabit savannah. Thus, the anthrax disease, considered a natural disease of ungulates, from 2010 is also a primate disease caused by *Bacillus cereus* biovar *anthracis*. *B. cereus* biovar *anthracis* is different from the pathogenic *B. cereus* strains discussed above that inherited a plasmid which enabled them to behave as anthrax-like pathogens and retain fully functional *B. cereus* PlcR-PapR quorum sensing. In *B. cereus* biovar *anthracis*, the PlcR-PapR quorum sensing is not functional, as in *B. anthracis*. *Bacillus cereus* biovar *anthracis* is the first isolate in which *B. anthracis* virulence plasmids are present in a non- *B. anthracis* chromosomal background (pBCXO1 and pBCXO2 with 99–100% of identity with *B. anthracis* pXO1 and pXO2 plasmids), while other pathogenic *B. cereus* strains contain only pXO1-like plasmids. *B. cereus* biovar *anthracis* also expresses the polyglutamate capsule coded by the pBCXO2 plasmid. *Bacillus cereus* biovar *anthracis* deserves special attention as a pathogen because in one plasmid, pBCXO1, reside both the determinants of pathogenicity (toxin and capsule), while *B. anthracis*, to be pathogenic, needs pXO1 (tripartite toxin) and pXO2 (polyglutamate capsule) plasmids. Indeed, *B. anthracis* is not pathogenic without pXO2 plasmid because this plasmid codes for the polyglutamate capsule that allows the bacterium to evade the immune system; in fact, *B. anthracis* Sterne strain, which lacks the pXO2 plasmid, is used as a live vaccine for livestock [38]. On the contrary, *B. cereus* biovar *anthracis* strain retains its pathogenicity without pBCXO2, because the pBCXO1 codes both for the tripartite toxin and for the hyaluronic acid capsule, co-ordinately expressed by the transcription factor AtxA [39]. In *B. anthracis* strains the hyaluronic acid capsule is not expressed because of a mutation in the has ACB operon. The expression of a hyaluronic acid capsule offers an advantage to bacteria during the respiratory tract infection, as happens for *Streptococcus pyogenes*, which is able to evade the host immune system [40,41]; this feature could facilitate the development of the pulmonary anthrax disease. *B. cereus* biovar *anthracis* is motile, while *B. anthracis* chromosome contains 10 mutations in chemotaxis and motility genes; the flagellum is an additional determinant of pathogenicity [42]. These new features (primates as a host, one plasmid highly pathogenic, a hyaluronic acid capsule adapted for respiratory tract infection, the illness caused is pulmonary anthrax) make *B. cereus* biovar *anthracis* strain potentially even more dangerous than *B. anthracis* for humans. Even if no infections have been observed in humans, specific antibodies against *Bacillus cereus* biovar *anthracis* have been detected in residents of the region, where this strain was isolated in Côte d’Ivoire [43]. This example underlines the necessity to have a close up look to these strains to prevent a possible human pathogenic threat. For this reason, in 2018, *Bacillus cereus* biovar *anthracis* was added to the list of biological agents in US and to the warning list of human and animal pathogens and toxins of the Australia Group Regime, an informal association of member states that aims to co-ordinate national export control laws to minimize the risk of proliferation of chemical and biological weapons.

We already have considerable knowledge of the cellular biology and pathogenicity of bacilli in the context of their terrestrial ecology but know little about the capabilities of bacilli in relation to their extraterrestrial ecology. Here, we report the first observation of a natural (i.e., not based on a scientific experiment) evolution of *Bacillus* strains on the ISS. We identify the likely routes by which bacilli escape the Earth during their invasion of space habitats and we consider the ecology of bacilli in space with respect to the astronaut microbiome. Finally, we consider the vulnerability of humans during space missions to outbreaks of anthrax and related infections.

## 2. Materials and Methods

### 2.1. Phylogenetic Analysis of ISS Bacillus Isolates Relative to Other Bacilli

The phylogenetic analysis of the ISS *Bacillus* strains was performed using the PathoSystems Resource Integration Center (PATRIC), a bacterial Bioinformatics Resource Center (https://www.patricbrc.org accessed on 15 December 2020) [44]. For this analysis, the PATRIC Codon Trees service uses up to 100 genome entry, thus, in addition to the 11 ISS strains, we select 85 *B. cereus sensu lato* complete genomes publicly available, from the PATRIC tree (https://www.patricbrc.org/view/Taxonomy/1386#view_tab=phylogeny, accessed on 20 December 2020) and using the *Geobacillus* group as the outgroup (a phylogenetic group distant from the *Bacillus cereus* group). The list of 96 genomes used for the phylogenetic analysis of Figure 2 is reported in Appendix A. For this comparison, the PATRIC Codon Trees service was used: the Codon Tree method selects single-copy of PATRIC’s global Protein Families (PGFams) [45], which are selected randomly and aligned using MUSCLE [46] and the coding gene sequences are aligned using the Codon_align function in BioPython [47]. The alignments are written in a phylip formatted file and a RAxML is generated [48]. Support values are produced through 100 rounds of the “Rapid” bootstrapping [49], option of RaxML. This service generates a newick phylogenetic tree basing on the differences within those selected 100 genes translated into proteins to draw the evolutionary tree. The list of proteins that this tool used for this analysis is listed in Appendix A. Analysis of Genetic Relatedness between ISS Isolates.

The phylogenetic analysis of the ISS strains (Figure 3) compared with four reference genomes (one representative for each specie, *B. cereus*, *B. anthracis*, *B. thuringiensis* and *B. cereus* biovar *anthracis*, the accession numbers of the genome sequences are reported in Appendix A) was performed with the PATRIC Codon Trees service, described above, selecting 100 proteins (listed in Appendix A).

### 2.2. Differences between Genomes and Predicted Protein Products of ISS Isolates

To compare the ISS *Bacillus* genomes, the PATRIC RAST Sequence-based Comparison tool was used [50]. These results were processed by the Proteome Comparison service of PATRIC Bioinformatics Resource Center to compare proteome profiles across different ISS species with *B. anthracis* str. Ames as reference (the row data are available upon request). The genomic sequence formats of two strains (Bacillus ISSFR-3F and Bacillus S1-R5C1-FB) were not suitable for the analysis performed by PATRIC and were not included in this analysis. This tool defines colors for each gene based on protein similarity using BLASTP when compared to the reference genome (Figure 5).

### 2.3. Synteny-Matrix (Dot-Plot) Analyses of Nucleotide Sequences

The synteny matrix analysis showed in Figures 6 and 7 was performed using the BLAST resource of the National Center for Biotechnology Information (NCBI, https://www.ncbi.nlm.nih.gov accessed on 1 November 2022). The BLAST program was used to align the *B. anthracis* str. Ames reference genome with the following ISS strains: Bacillus ISSFR-3F, Bacillus ISSFR-9F, Bacillus JEM-2, chosen because they are (among 11 ISS strains) the only strains with the complete chromosomal and plasmid sequences available on the NCBI database (GeneBank accession numbers: *B. anthracis* str. Ames: AE016879.1. Bacillus ISSFR-3F: CP018931.1. Bacillus ISSFR-9F: CP018933.1. Bacillus JEM-2: CP018935.1). The Dot Plot analysis was chosen to visualize the regions of similarity based on the BLAST results. The *B. anthracis* str. Ames is represented on the X-axis; for each plot, an ISS strain is represented on the Y-axis. The numbers represent the bases/residues. Alignments are shown in the plot as lines. Plus strand and protein matches are drawn from the bottom left to the upper right, minus strand matches are drawn from the upper left to the lower right.

## 3. Results and Discussion

### 3.1. The ISS Bacillus Strains Are Close Related to B. cereus Biovar Anthracis and to B. cereus Human Pathogenic Strains

Even if the primary habitats of the aerobic endospore-forming bacilli are soils, these bacteria are found everywhere including plants, brines, the Earth’s subsurface and atmosphere, and in anthropogenic environments, including spacecraft. A microbiological survey on dust collected in the ISS, identified 6 *Bacillus* strains from the Japanese (two strains) and the U.S. modules (four strains) [51]; another study reported the isolation of 5 *Bacillus* strains from the Russian module [52,53] (Figure 1); from these isolates, the genomes of 11 strains (from now on called ISS *Bacillus* strains) were sequenced because the initial screening led to the suspicion that these strains could be *B. anthracis* [54].

The *Bacillus* species discussed in this paper were isolated from:the U.S. segment Harmony Node 2, from air HEPA filters used 40 months (returned with flight STS-134/ULF6, 2011);the Kibo Japanese experimental module, from air diffuser samples collected with a surface sample kit (Expedition 19, 2009);the Russian segment Zvezda Service Module (DOS-8), from surface samples collected with a Swab Rinse Kit tube (ESA Delta mission expedition 8, 2004 and expedition 11, 2005).

The phenotypic and sequence analysis identified them as members of the *B. cereus* group but excluded they could be *B. anthracis* because they lacked the pXO1 and pXO2 plasmids; nevertheless, the DNA-DNA hybridization analysis revealed that the ISS isolates were similar to *B. anthracis* but distant to *B. cereus* and *B. thuringiensis* strains. These isolates form a new clade distinct from other strains of the *B. cereus sensu lato* group but closely related to *B. anthracis* [54].

Thus, in the ISS a new group of *B. cereus* were detected, not really *B. cereus*, nor really *B. anthracis*. So, the key question is, are astronauts safe from anthrax-like infection from these bacilli? The answer to this question is no; indeed, they are at risk. The phylogenetic tree shows the ISS *Bacillus* strains close related to *B. cereus* biovar *anthracis* and to *B. cereus* human pathogenic strains (Figure 2).

The motility exhibited by the ISS *Bacillus* strains is a *B. cereus* feature [54], but it is noteworthy that also *B. cereus* biovar *anthracis* is motile [35], conferring an advantage to the bacterium for host colonization. Furthermore, motility can greatly enhance competitive ability against other micro-organisms [55], a factor that may increase the chances of host infection. The ISS *Bacillus* strains have a functional PlcR transcription regulator, essential for the efficient adaptation of *B. cereus* strains to the environment; the PlcR transcription factor controls also the expression of most known virulence factors [14].

Whether *B. cereus* could become more virulent in the space has not been demonstrated/observed, but the changes in metabolism, growth rate and resistance to antibiotics were influenced by the space environment in a *B. cereus* strain that was sent to space by the ShenZhou VIII spacecraft, from 1–17 November 2011 [56].

Successful detection of human pathogens in long-duration human missions is essential, together with mitigation protocols to eradicate pathogens identified in space. The spores are the main concern because they are resistant to desiccation, heat, radiation and disinfectants and can be identified in dusts and aerosols, but they can be inactivated by cold plasma [57]. In addition, hygiene regimes that make use of cleaning solutions containing disinfectants may clean selectively, favoring the selection of spore-forming *Bacillus* strains that possess features to cope with hostile environments. Experiments performed on board of the NASA Long Duration Exposure Facility showed that *B. subtilis* spores were still viable after the exposition to the space for six years [58] and *B. subtilis* spores could survive in a Mars surface environment, if shielded from UV by dust or rock [59]. Indeed, spore-forming bacterial species, such as of the genus *Bacillus*, have been detected on the ISS in considerable numbers. A microbiological survey over a period of 6 years identified *Bacillus* as the second most encountered genus (31.7%) [60]. It is of interest that in the Russian module a microbiological survey along six years recovered *B. cereus* among six *Bacillus* species [60]. In 2014, a *B. thuringiensis* strain was identified in the Russian module with an experiment had the aim to test 4 materials, aluminum and polymers, as a substrate for microbial growth. These materials were exposed on the ISS over a period of 135 days [61]. Recently, metagenome sequence analysis from eight ISS environmental sites revealed the presence of microbial communities with antimicrobial resistance profiles, and virulence properties, among them *B. anthracis* [62]. Thus, the *Bacillus* cereus group are not uncommon in the ISS. Moreover, from dust particles collected at the Jet Propulsion Laboratory Spacecraft Assembly Facility (US), another new *Bacillus* specie was isolated, *Bacillus nealsonii*, whose spores are resistant to UV, gamma radiation, H_2_O_2_ and desiccation [63]. This is of interest because some cargo sent to the ISS has been packaged in these cleanrooms. Indeed, with a microbiological survey of the commercial resupply vehicle, *Bacillus* strains were commonly identified, and it is mandatory to assess the potential risk of transfer micro-organisms from the Earth to the ISS [64].

How did the ISS *Bacillus* strains arrive in the ISS and spread along the three modules? The ISS length of pressurized modules is 73 meters and the astronauts (no more than 6) and materials move along the modules, facilitating the movement of micro-organisms. Unfortunately, the microbiological survey experiments were performed in different times and with different methods of sampling, and it is unknown if those strains were living in a vegetative form, or they were in a quiescent state as spores. Nevertheless, the comparison of their genomic sequences allowed us to formulate a hypothesis. The phylogenetic tree of the ISS *Bacillus* strains reveals that the ancestral strains are those isolated from the US module and that the strains isolated from the Russian and Japanese modules were derived from the US strains (Figure 3).

Even if the genetic differentiation between these strains is low, this result could indicate that, from the arrival to the ISS and the sampling, they were in a vegetative form for a period sufficient to accumulate genetic variations. We can hypothesize that a *Bacillus cereus* strain arrived in the ISS before 2004 (the year of the first sampling), spread along the ISS, remained as spore in the US module and was captured by the HEPA filters in 2011, while in the Russian and Japanese modules it remained in the vegetative form long enough to fix a few genetic variations. An alternative hypothesis is that the ISS *Bacillus* strains arrived in the ISS in different periods, always from the same terrestrial contamination source.

### 3.2. Natural Evolution of Bacilli in the ISS

In any way they arrived, those strains were in the ISS many years but, most important, they are still there, and they are evolving. Hence, the necessity to identify the molecular differences among the ISS *Bacillus* strains more closely. We compared the whole genome sequences of the ISS *Bacillus* strains with *B. anthracis* strain Ames (chosen as a reference strain because the *Bacillus* ISS clade is related to the *B. anthracis* clade) with the PATRIC tool (see Materials and Methods). At nucleotide level, the whole genome alignments revealed a total of 40,902 variants (Single Nucleotide polymorphisms, SNPs, Insertions and deletions) that differentiate *B. anthracis* strain Ames from the 9 ISS *Bacillus* strains (Figure 4).

In total, 37,679 out of 40,902 of these variants were present in all the ISS *Bacillus* strains, suggesting that they derive from a single strain. The remaining 3223 variations, compared to *B. anthracis* str. Ames genome, were found to be unique in the ISS *Bacillus* strains or present in more than one strain. Of note, 1047 out of 3223 variations are shared by all but *Bacillus* ISSFR-25F. This observation could suggest that this strain has diverged earlier from the others (see also Figure 3); indeed, *Bacillus* ISSFR-25F showed the highest number of unique variations, 287, i.e., not present in other strains. Unique genetic variants of the ISS *Bacillus* strain/s compared with *B. anthracis* str. Ames, with a predicted high impact on protein product are listed in Appendix A. We must underline that comparing the ISS *Bacillus* genomes with *B. anthracis*, our aim was to enlighten the differences among the ISS *Bacillus* strains to show the genome evolution on the ISS, indeed, many variants are shared with terrestrial *B. cereus* strains.

At protein level, the percentage of protein identity of the *Bacillus* ISS predicted proteins compared with that of *B. anthracis* showed that the identity has a range between 95 and 100% (Figure 5) with only a few regions with less identity, approximately 70%. Interestingly, the ISS strains lack the four *B. anthracis*-specific prophage regions [65].

Of note, the *Bacillus* ISS strains, from circle 2 to 10, are quite identical at protein level, except for the ISS *Bacillus* ISSFR-23F strain, distinct from those of other ISS *Bacillus* strain and *B. anthracis* strain, showing 4 genomic regions with low identity (marked with black arrows in Figure 5). The ISSFR-23F strain is also characterized by the lack of 19 proteins, present in all the other ISS *Bacillus* strains. The detailed list of proteins lacking at least in one ISS *Bacillus* strain, compared with *B. anthracis* str. Ames, are listed in Appendix A. The loss of some genes only in ISSFR-23F and the divergence in 4 genomic regions, clearly suggests an evolution of the genomes on the ISS, and since the proteins of the diverged regions are identical to *B. cereus* proteins present in databases, we cannot exclude that a recombination occurred among the ISS *Bacillus* ISSFR-23F with other *B. cereus* strains present in the ISS.

Rearrangements of large portion of genomes is common in bacterial evolution, and such variations cannot be identified with the bioinformatic analysis described above, thus, a synteny matrix (dot-plot) analyses of the *Bacillus* ISS strains with *B. anthracis* str. Ames was performed using the BLAST resource of the National Center for Biotechnology Information. The alignments showed that the ISSFR-9F had a synteny with *B. anthracis* str. Ames, while the ISSFR-3F showed a genome inversion with respect to the replication origin, and the JEM-2 strain has a rearrangement of a large part of the genome (Figure 6A). A graphical representation of the genomic rearrangements is shown in Figure 6B. At genomic level, this result confirms an ongoing evolution of the ISS *Bacillus* strains.

### 3.3. ISS Bacillus Plasmids

All the ISS strains contained a plasmid similar to the pBT9727 plasmid of *B. thuringiensis* serovar *konkukian* strain 97-27 [66]. This *B. thuringiensis* strain was isolated from a case of severe human tissue necrosis infection [28] and harbours the pBT9727 conjugative plasmid; it is able to promote its own transfer to other bacteria and shares a common ancestor with *B. anthracis* pOX2 [67]: the pBT9727 is not pathogenic because the pXO2 region encoding a polyglutamic acid capsule is replaced by genetic mobile elements [68]. We performed synteny matrix (dot-plot) analyses of the *Bacillus* ISS plasmids with the *B. thuringiensis* serovar *konkukian* 97-27 plasmid pBT9727 (Figure 7).

In accordance with the plasmid sequences present in GeneBank, the ISS *Bacillus* plasmids from ISSFR-3F, ISSFR-9F, JEM-2 and ISSFR-23F showed a conserved synteny, but the ISSFR-9F plasmid is inverted with respect to the replication origin. All plasmids contained duplicated regions and showed a deleted region containing a hypothetical IS231 transposase in pBT9727. Being also the chromosomal genome of *B. thuringiensis* serovar *konkukian* 97-27 close related to the ISS *Bacillus* genomes (Figure 2), we could speculate that a *B. thuringiensis* serovar *konkukian* 97-27 related strain could be the progenitor of the ISS *Bacillus* strains. Nevertheless, a comparison with the BLAST program showed the ISS *Bacillus* genomes shared higher identity with *B. anthracis* sp. Ames than with *B. thuringiensis* serovar *konkukian* 97-27 (not shown). Based on our results, we favor the simplest hypothesis that a *B. cereus* strain, harboring a *B. thuringiensis* serovar *konkukian* 97-27 related plasmid, arrived in the ISS before 2004 (the year of the first sample) and it spread along the ISS. We cannot exclude that the ISS *Bacillus* ancestor acquired the plasmid from another *Bacillus* strain present on the ISS.

The ISS *Bacillus* ISSFR-25F strain separated early from the others because it accumulated many variations (287) not shared by other strains, while the remaining strains (ISSFR-23F, JEM-2, ISSFR-9F, JEM-1, S1-R4H1-FB, S2-R3J1-FB-BA1, S1-R1J2-FB, S1-R2T1-FB) all share the same 1051 genetic variations. The presence of genetic variations among these strains suggests they have spent at least a period in the ISS as a vegetative form, permitting the acquisition of genetic variants during DNA replication. The observation that these *Bacillus* strains could live in a vegetative form is a concern because we speculate that these strains are close to *B. anthracis* strains and *B. cereus* human pathogenic strains, and they could exchange or eventually acquire pathogenic plasmids from other members of the *B. cereus* group, already present in the ISS. Of note, strains of *B. cereus* and *B. thuringiensis* were already detected in the ISS [60,61] but, unfortunately, if these strains harbored plasmids was not reported by the authors. Thus, a new clade of the *Bacillus cereus* group, close related to *B. anthracis*, colonized the entire ISS and it is still there.

## 4. Conclusions

Which kind of treat deriving from the *Bacillus cereus* group should be evaluated for the astronauts? The contamination from biological experiments carried out in the ISS is low because highly infective or pathogenic biological materials are not allowed on any payload experiment. Infection from a natural strain? It could be possible, because we showed the new clade of *B. cereus* emerged from the ISS is close to *B. anthracis* strains and *B. cereus* and *B. thuringiensis* human pathogenic strains. *Bacillus* strains are among the most resistant bacteria and can colonize every substrate; once they arrive on the ISS, on the Moon or on Mars it will be difficult to eradicate them. Most of the microbiological studies are based on surveys on microbiome of the ISS to monitor the presence of human pathogens, giving a snapshot of the presence of micro-organisms in a specific moment; as an example, *B. cereus* and *Staphylococcus aureus* pangenomes from the ISS were compared with terrestrial isolated from both built environments and humans [69]. Nevertheless, the strains detected onboard are still onboard of the ISS and can be a potential threat in relation to accidental infection (biosafety) and intentional infection (biosecurity/bioterrorism). The possibility of a bioterrorist attack is extremely low due to the tied controls and the astronauts are vaccinated for diphtheria and tetanus, polio, hepatitis A, hepatitis B, measles/mumps/and rubella, varicella; moreover, once a year, they are tested for tuberculosis. However, are they also vaccinated for anthrax?

The discovery of *B. cereus*-group strains in the ISS should increase awareness of biosafety and biosecurity risks because we demonstrate they are evolving, showing genomic and plasmid plasticity.

## Figures and Tables

**Figure 1 microorganisms-10-02309-f001:**
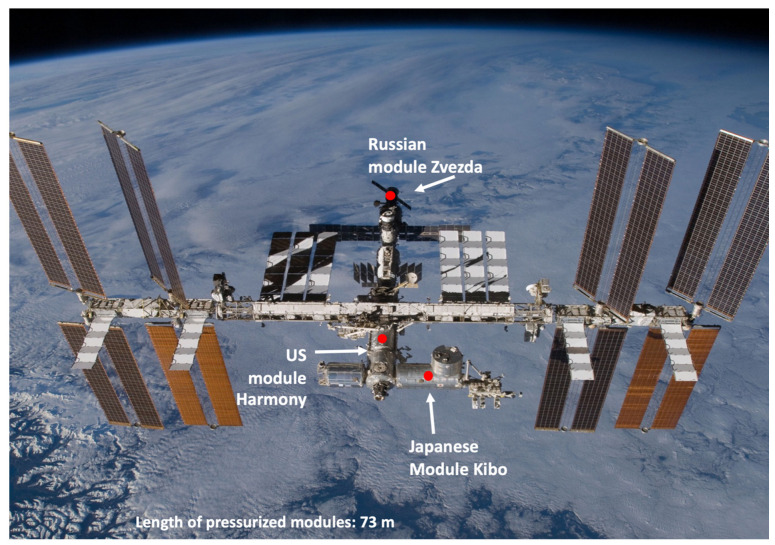
International Space Station in which the Russian, US and Japanese modules are indicated (red dots). In these modules *Bacillus* strains were recovered during microbiological surveys. Credit: NASA, https://www.nasa.gov accessed on 1 December 2020.

**Figure 2 microorganisms-10-02309-f002:**
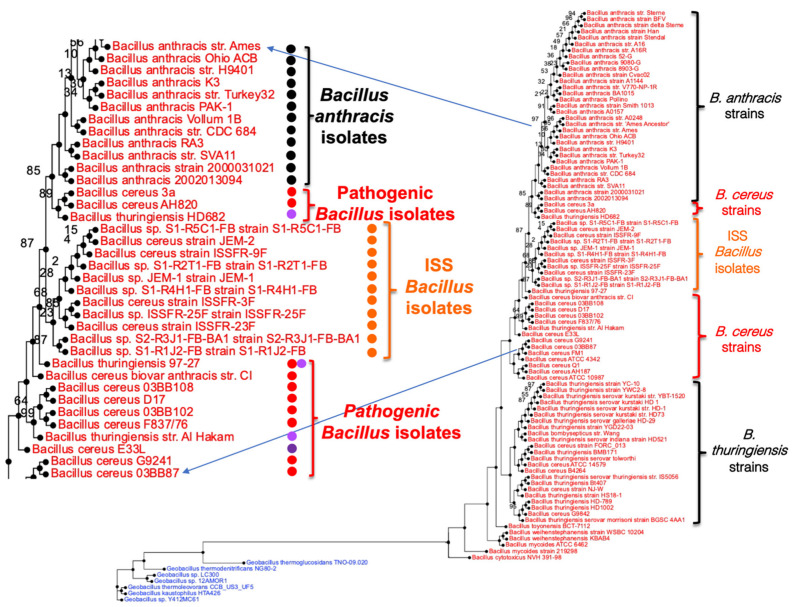
The ISS *Bacillus* isolates are close related to B. *anthracis* and B. *cereus* pathogenic strains. Maximum-likelihood phylogenetic tree was constructed using complete genome sequences of 96 *Bacillus* isolates (listed in SI) using the PathoSystems Resource Integration Center (PATRIC) (see M&M). *Bacillus* isolates obtained from the ISS between 2005 and 2011 are indicated by orange dots and orange font. Terrestrial *Bacillus* isolates known to be pathogenic to humans and/or other primates (causing anthrax like disease) are indicated by red dots and red font. Terrestrial *Bacillus* isolate pathogenic to other animals is indicated by purple dot. Those isolated obtained from other courses (e.g., soil) are indicated by black dots and black font. Terrestrial *B. thuringiensis* strains lacking genes coding for the parasporal crystal are indicated by pale purple dots (for references, see Appendix A).

**Figure 3 microorganisms-10-02309-f003:**
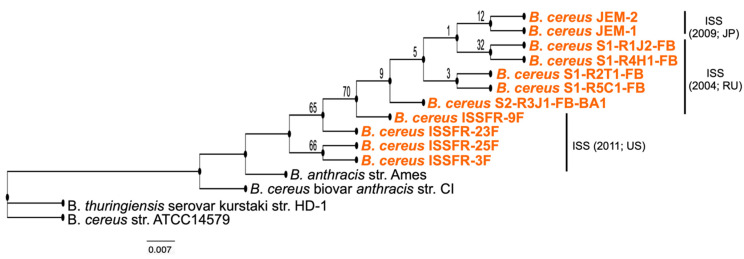
ISS *Bacillus* strains are closely related to the pathogenic Ames *B. anthracis* strain ISS strains and very closely related to each other. Maximum-likelihood phylogenetic tree was constructed using the PathoSystems Resource Integration Center (PATRIC) using complete genome sequences of 11 ISS *Bacillus* isolates sampled in the ISS between 2005 and 2011 (indicated by orange font) and four terrestrial *Bacillus* isolates for comparison. *B. anthracis* str. Ames and *B*. *cereus* biovar *anthracis* str. CI are pathogenic strains.

**Figure 4 microorganisms-10-02309-f004:**
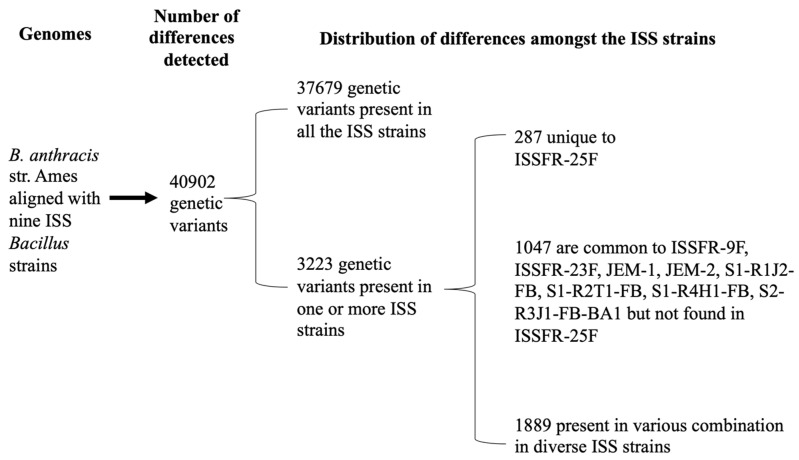
Number of genetic variants (point mutations, recombinations and/or deletions) between whole genomes of ISS *Bacillus* isolates (ISSFR-9F, ISSFR-23F, ISSFR-25F, JEM-1, JEM-2, S1-R1J2-FB, S1-R2T1-FB, S1-R4H1-FB, S2-R3J1-FB-BA1) and *B. anthracis* str. Ames. Unique genetic variants of the ISS *Bacillus* isolates, with a predicted high impact on protein product, are reported in Appendix A.

**Figure 5 microorganisms-10-02309-f005:**
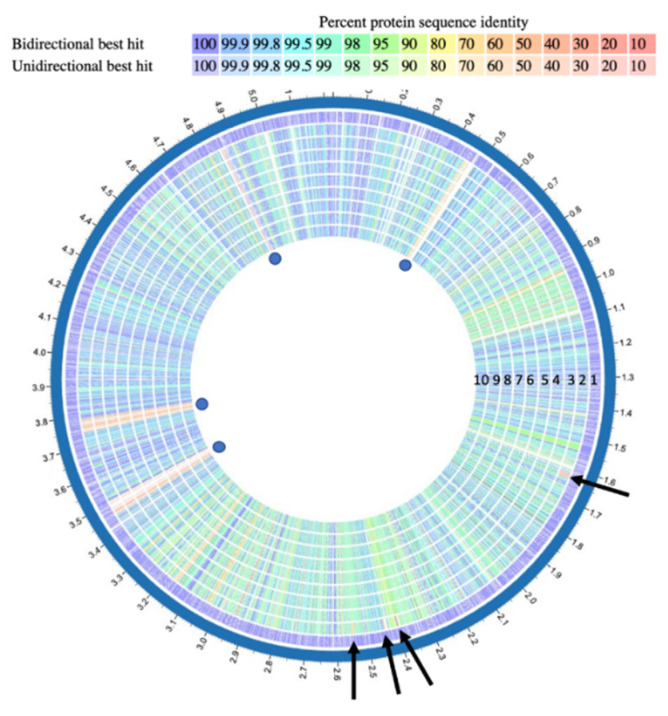
The chromosome-encoded proteins of ISS *Bacillus* isolates compared with those of *B. anthracis* str. Ames. The *B. anthracis* str. Ames proteins, circle 1, are depicted in purple, this colour indicates 100% identity. Only a few regions of the ISS *Bacillus* strains are below the 60–70% of identity, in peach colour, compared with *B. anthracis* str. Ames, the phage regions absent in the ISS strains are marked with blue dots. The plot also shows the high identity of the ISS *Bacillus* proteins, with the exception of *Bacillus* ISSFR-23F strain, circle 2, which shows divergence in some protein regions, black arrows. The list of proteins lacking at least in one ISS *Bacillus* strain compared with B. *anthracis* str. Ames, are reported in Appendix A. Genomes, from outside to inside: circle 1: B. *anthracis* str. Ames (purple); circle 2: *Bacillus* sp. ISSFR-23F; circle 3: *Bacillus* sp. JEM-2; circle 4: *Bacillus* sp. ISSFR-9F; circle 5: *Bacillus* sp. JEM-1; circle 6: *Bacillus* sp. S1-R4H1-FB; circle 7: *Bacillus* sp. S2-R3J1-FB-BA1; circle 8: *Bacillus* sp. S1-R1J2-FB; circle 9: *Bacillus* sp. ISSFR-25F; circle 10: *Bacillus* sp. S1-R2T1-FB. Blue dots: phage regions absent in the ISS strains. Black arrows: regions of the *Bacillus* ISSFR-23F strain divergent from the other *Bacillus* ISS strains.

**Figure 6 microorganisms-10-02309-f006:**
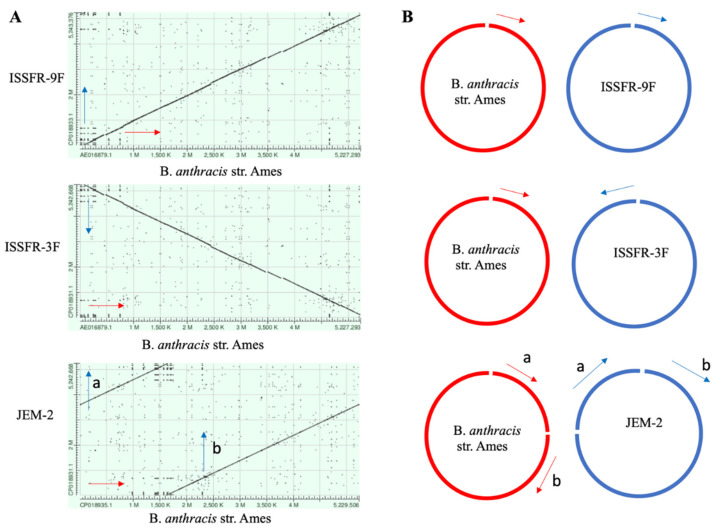
Synteny matrix (dot-plot) analyses of the *Bacillus* ISS strains with *B. anthracis* str. Ames. (**A**) Dot Plot representations (BLAST score ratio-based synteny plots) of alignments between B. *anthracis* str. Ames chromosome (X-axis) and the ISS *Bacillus* strains (Y-axis). Each point on the figure represents an individual peptide in B. *anthracis* str. Ames genome compared to the proteome of the ISS strains. (B) Graphic representation of the plots showed in (A). The ISSFR-3F strain showed the same orientation of the B. *anthracis* str. Ames. The ISSFR-3F chromosome show a reverse relative orientation compared with the B. *anthracis* str. Ames chromosome. The JEM-2 chromosome shows a translocation: the chromosome segment indicated with (a) in B. *anthracis* str. Ames corresponding to the first part of the chromosome, is found in the last part in JEM-2. GeneBank accession numbers: B. *anthracis* str. Ames AE016879.1, 5,227,293 bp. *Bacillus* ISSFR-3F CP018931.1, 5,242,668 bp. *Bacillus* ISSFR-9F CP018933.1, 5,243,376 bp. *Bacillus* JEM-2 CP018935.1, 5,229,506 bp.

**Figure 7 microorganisms-10-02309-f007:**
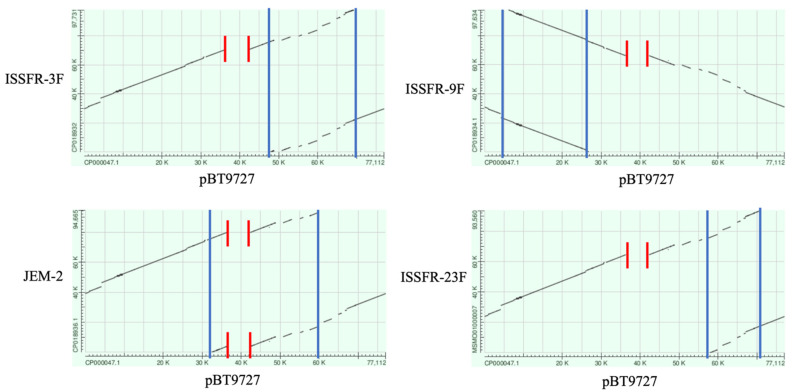
Dot Plot representations (BLAST score ratio-based synteny plots) of alignments between *B. thuringiensis* serovar *konkukian* 97-27 plasmid pBT9727 (X-axis) and the ISS Bacillus strains (Y-axis, ISSFR-3F, ISSFR-9F, JEM-2 and ISSFR-23F from top left to bottom right, respectively). Each point on the figure represents an individual peptide in pBT9727 compared to the proteome of the ISS plasmids. Red lines indicate the region containing a hypothetical IS231 transposase in pBT9727 absent in the ISS Bacillus plasmids. Based on the sequences present in GeneBank, the plasmid regions between the Blu lines are duplicated regions in the ISS plasmids. The ISSFR-9F plasmid shows a reverse relative orientation compared with pBT9727. GeneBank accession numbers: *B. thuringiensis* serovar *konkukian* pBT9727 plasmid CP000047.1, 77112 bp. *Bacillus* ISSFR-3F plasmid CP018932, 97,731 bp. *Bacillus* ISSFR-9F plasmid CP018934.1, 97,634 bp. *Bacillus* JEM-2 plasmid CP018936.1, 94,665 bp. *Bacillus* ISSFR-23F plasmid MSMO01000007, 93,560 bp.

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
