# Peer review of "Bacilli in the International Space Station"

_microorganisms, 2022, doi:10.3390/microorganisms10122309_

Round 1

Reviewer 1 Report

Manuscript Quagliariello et al focuses on a poorly explored aspect of the biology of Bacilli and reports a genomic characterization of B. cereus strains isolated from the International Space Station (ISS). These 11 strains were classified as B. cereus but are closely related to B. anthracis. The genomic analysis allowed the Authors to suggest that all ISS strains originated from a single strain that contaminated that environment spreading through the ISS and accumulating mutations. However, in order to accumulate all the identified mutations cells need to go through a huge number of cell divisions. Is this possible in the ISS environment? Considering the frequency of mutation, how many cell divisions should occur to originate all the variations observed? Has been previously reported that the frequency of mutation is higher in the ISS than in a terrestrial environment?

Other comments:

1) it is not easy to discriminate original from literature data. I suggest to split Results from Discussion and report in the Results section only the original data

2) The phenotypic analysis (line 209) and the cell motility (line 231) were experimentally tested or reported in previous studies?

3) lines 209-214: it is not clear to me if these sentences refer to literature data (but there are not references) or experiments performed inthis manuscript (in this case, please give details)

4) lines 324-326: since prophage DNA lacks in all ISS strains seems likely that the original ccontaminant also lacked these regions. Is this correct or the Authors suggest that the original strain may have lost these regions in the ISS?

5) were these 11 strains isolated as spores or vegetative cells (were samples heat treated before plating?)

Author Response

Ref 1

Manuscript Quagliariello et al focuses on a poorly explored aspect of the biology of Bacilli and reports a genomic characterization of B. cereus strains isolated from the International Space Station (ISS). These 11 strains were classified as B. cereus but are closely related to B. anthracis. The genomic analysis allowed the Authors to suggest that all ISS strains originated from a single strain that contaminated that environment spreading through the ISS and accumulating mutations. However, in order to accumulate all the identified mutations cells need to go through a huge number of cell divisions. Is this possible in the ISS environment? Considering the frequency of mutation, how many cell divisions should occur to originate all the variations observed? Has been previously reported that the frequency of mutation is higher in the ISS than in a terrestrial environment?

Other comments:

1) it is not easy to discriminate original from literature data. I suggest to split Results from Discussion and report in the Results section only the original data.

Thank you for the comment, unfortunately to organize and split the section will take more than the time we have to send back the manuscript with comments, we added the references at the end of the sentences to facilitate the readers to distinguish among our data and the literature.

2) The phenotypic analysis (line 209) and the cell motility (line 231) were experimentally tested or reported in previous studies?

Thank you for the question, yes, it was experimentally tested, and we added the appropriate references in the sentence.

3) lines 209-214: it is not clear to me if these sentences refer to literature data (but there are not references) or experiments performed in this manuscript (in this case, please give details)

Thank you, we added the corresponding reference in the text.

4) lines 324-326: since prophage DNA lacks in all ISS strains seems likely that the original contaminant also lacked these regions. Is this correct or the Authors suggest that the original strain may have lost these regions in the ISS?

 Thank you for the observation, we clarify the sentence and added a reference. The 4 prophage regions are always present in B.anthracis strains, while they are generally absent in B.cereus strains. It is highly probable the ISS progenitor strain already lacked these regions.

5) were these 11 strains isolated as spores or vegetative cells (were samples heat treated before plating?)

Thank you for the interesting question, the samples were not heat treated, thus the strains could have been in a vegetative or in a spore form at the time of sampling. That’s the reason why it is very difficult to trace the evolution of those strains.

Reviewer 2 Report

The idea of collecting bacilli from the International Space Station to study their evolutionary biology and pathogenicity is a novel but realistic idea. Certainly, this research has yielded some meaningful results. In summary, such research is intriguing and the choice of this journal is appropriate.

The phylogenetic tree analysis is the most common research method used in this manuscript, however, the author only mentions the algorithm of Maximum-likelihood, and the software used and its related parameters (such as bootstrap) need to be specified.

There are many detailed problems in the references, the abbreviation-full name of the journal is not uniform, the page number format is inconsistent,

Also, a non-professional question, Figure 1, is the top view of the ISS authorized, and how did it come about.

Author Response

The idea of collecting bacilli from the International Space Station to study their evolutionary biology and pathogenicity is a novel but realistic idea. Certainly, this research has yielded some meaningful results. In summary, such research is intriguing and the choice of this journal is appropriate.

The phylogenetic tree analysis is the most common research method used in this manuscript, however, the author only mentions the algorithm of Maximum-likelihood, and the software used and its related parameters (such as bootstrap) need to be specified.

 Thank you for the comment, we have clarified this point in the text.

There are many detailed problems in the references, the abbreviation-full name of the journal is not uniform, the page number format is inconsistent

We have fix this problem, thank you

Also, a non-professional question, Figure 1, is the top view of the ISS authorized, and how did it come about.

Thank you, we have fixed this point, the credit is NASA.